# Pre-Launch Radiometric Characterization of EMI-2 on the GaoFen-5 Series of Satellites

**Minjie Zhao** [1] , **Fuqi Si** [1,*], **Haijin Zhou** [1] , **Yu Jiang** [1] , **Chunyan Ji** [1,2], **Shimei Wang** [1] , **Kai Zhan** [1] **and Wenqing Liu** [1]

1   Key Laboratory of Environmental Optics and Technology, Anhui Institute of Optics and Fine Mechanics, Chinese Academy of Sciences, Hefei 230031, China; mjzhao@aiofm.ac.cn (M.Z.); hjzhou@aiofm.ac.cn (H.Z.); yjiang@aiofm.ac.cn (Y.J.); cyji62@mail.ustc.edu.cn (C.J.); wsm@aiofm.ac.cn (S.W.); zhankai@aiofm.ac.cn (K.Z.); wqliu@aiofm.ac.cn (W.L.)
2   University of Science and Technology of China, Hefei 230026, China
*   Correspondence: sifuqi@aiofm.ac.cn

**Abstract:** The environmental trace gas monitoring instrument (EMI) is a space-borne imaging spectrometer onboard GaoFen-5, which was launched in May 2018, covering wavelengths in the range of 240–710 nm to measure $NO_2$, $O_3$, HCHO, and $SO_2$. An advanced EMI-2 instrument with a higher spatial resolution and sufficient signal-to-noise is currently planned for launch on the GaoFen-5(02) satellite in 2021. The EMI-2 instrument bidirectional scattering distribution function (BSDF) is obtained from the absolute irradiance and radiance calibration on-ground. Based on EMI-2 earth and sun optical paths, the key factors of BSDF parameters are introduced. An NIST-calibrated 1000 W FEL quartz tungsten halogen lamp and a 2D turntable are adopted for the absolute irradiance calibration. A large aperture integrating sphere system is used for the absolute radiance calibration. Based on absolute irradiance and radiance calibration functions, the BSDF parameters are obtained, with accuracy of 4.9% for UV1, 4.3% for UV2, 4.1% for VIS1, and 4.2% for VIS2. The on-ground measurement results show that the reflectance spectrum can be calculated from BSDF parameters. On-orbit application of the EMI-2 instrument BSDF are also discussed.

**Keywords:** EMI-2; GaoFen-5; pre-launch; radiometric calibration; instrument BSDF





## 1. Introduction

The environmental trace gas monitoring instrument (EMI) onboard GaoFen-5 was launched in May 2018, its on-orbit performance was discussed in [1], and the trace gas products ($NO_2$, $O_3$, and $SO_2$) were introduced in [2–5]. The Environmental trace gas monitoring instrument-2 (EMI-2) is an improved version of EMI, which is a nadir-viewing push-broom imaging spectrometer, and is currently planned for launch on GaoFen-5(02) satellite in the time frame of 2021. EMI-2 and EMI have a descending node equator crossing time of 10:30 and an ascending node equator crossing time of 13:30, respectively. These two instruments have the same altitude of approximately 705 km and can be networked for remote sensing.

EMI-2 is a kind of differential optical absorption spectroscopy instrument similar to satellite instruments GOME-2 [6], OMI [7], and TROPOMI [8]. For retrieving atmospheric trace gas, EMI-2 detects atmospheric radiance in Earth mode and measures solar irradiance via onboard diffusers in Sun mode. The earth reflectance spectrum is obtained from the ratio of atmospheric radiance to solar irradiance, which are used as input for trace gas retrieval algorithms. The reflectance spectrum can be directly obtained from EMI-2 instrument bidirectional scattering distribution function (BSDF), which makes instrument BSDF an important radiometric calibration parameter. Instrument BSDF equals the ratio of EMI-2 radiance and irradiance calibration functions. For GOME, OMI, and TROPOMI, instrument BSDF is determined on the ground. The optical components are common to the radiance and irradiance optical paths; in principle, the GOME instrument BSDF is only

determined by the diffuser BSDF [9]. The OMI instrument BSDF is measured using an uncalibrated 300 W xenon high-pressure arc discharge lamp and an external spectralon plate with known BRDF [10]. The TROPOMI instrument BSDF is calculated from the calibration of absolute radiance and irradiance with absolutely calibrated FEL lamps and external calibrated diffusers [11]. For EMI, DOAS (Differential Optical Absorption Spectroscopy) technique [12] is adopted for trace gases retrieval, and the Sun mode is mainly used for in-flight wavelength calibration and for monitoring the optical degradation; therefore, EMI instrument BSDF has not been determined on ground. The EMI-2 instrument BSDF calibration is introduced in this paper to improve the earth reflectance spectrum calculated accuracy.

The EMI-2 optical design is almost a copy of the EMI concept, and the EMI detailed optical design was introduced in [13]. The major change between the two instruments is the telescope mirrors. The EMI-2 telescope adopts off-axis high-order aspheric mirror system instead of off-axis spherical mirror system to achieve a higher spatial resolution. The spatial resolution in the swath and flight directions is 13 km × 48 km for EMI and 13 km × 24 km for EMI-2. The optical layout of the EMI-2 is illustrated in Figure 1.

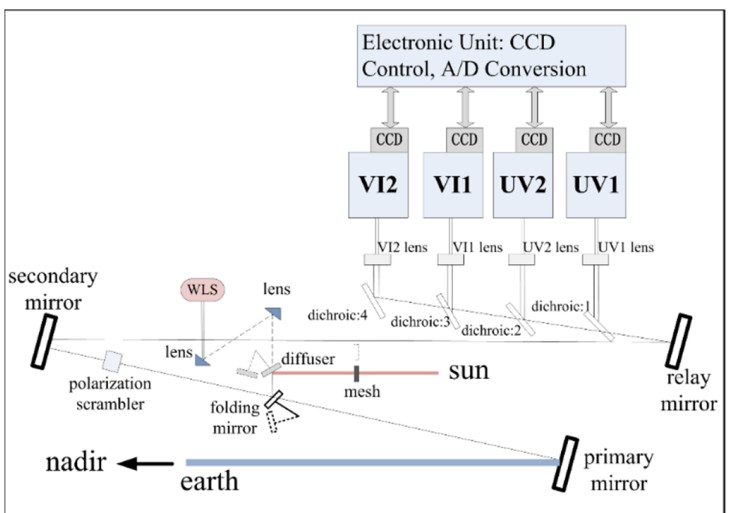

**Figure 1.** Optical layout of EMI-2.

EMI-2 has three observation ports: earth, sun, and white light source (WLS) port. It consists of a telescope and four Offner imaging spectrometers (namely, UV1, UV2, VI1, and VI2 channels), enabling an instantaneous field of view (IFOV) of 114° to realize one-day global coverage, ranging from 240 nm to 710 nm with a spectral resolution of 0.3–0.6 nm. Each channel adopts a 2D charge-coupled device (CCD) detector. One (along-track) dimension measures spectral information, and the other (across-track) dimension detects spatial information. A polarization scrambler makes EMI-2 insensitive to the polarization state of the incident light. Properties of EMI-2 are listed in Table 1.

The sun and earth have different optical paths: Solar radiation enters the telescope via onboard diffuser and folding and secondary mirrors. Atmospheric scattering light enters the telescope through primary and secondary mirrors. EMI-2 instrument BSDF represents the radiometric response relationship between two optical paths.

**Table 1.** EMI-2 instrument properties.

|  | EMI | EMI-2 |
|---|---|---|
| Spectral range | UV1: 240–315 nm, UV2: 311–403 nm; VI1: 401–550 nm, VIS2: 545–710 nm; | UV1: 240–311 nm, UV2: 311–401 nm; VI1: 401–550 nm, VIS2: 550–710 nm; |
| Spectral resolution | 0.3–0.5 nm | 0.3–0.6 nm |
| Telescope swath IFOV | 114° (2600 km on the ground) | 114° (2600 km on the ground) |
| Telescope flight IFOV | 0.5° (6.5 km on the ground) | 0.5° (6.5 km on the ground) |
| CCD detectors | UV: 1072 × 1032 (spectral × spatial) pixels VIS: 1286 × 576 (spectral × spatial) pixels | UV: 1072 × 1032 (spectral × spatial) pixels VIS: 1286 × 576 (spectral × spatial) pixels |
| Spatial resolution | 13 km × 48 km | 13 km × 24 km |
| Orbit | polar, sun-synchronous, ascending node equator crossing time: 13:30 | polar, sun-synchronous, descending node equator crossing time: 10:30 |

## 2. Methodology

EMI-2 on-orbit atmospheric radiance $L_{Earth\text{-}rad}(\lambda)[\mathrm{uW/cm^2/nm/sr}]$ and solar irradiance $E_{Sun\text{-}irrad}(\lambda)[\mathrm{uW/cm^2/nm}]$ are obtained as follows:

$$L_{Earth\text{-}rad}(\lambda) = f_{Rad}(\lambda) \cdot S_{Earth}(\lambda) \tag{1}$$

$$E_{Sun\text{-}irrad}(\lambda) = f_{Irrad}(\lambda) \cdot S_{Sun}(\lambda) \tag{2}$$

where $\lambda[\mathrm{nm}]$ is the wavelength of detector pixel; $S_{Earth}(\lambda)[\mathrm{DN}]$ and $S_{Sun}(\lambda)[\mathrm{DN}]$ are the output digital number of Earth and Sun modes, respectively, and corrected for dark signal, nonlinearity, and pixel response nonuniformity; $f_{Rad}(\lambda)[\mathrm{uW/cm^2/nm/sr/DN}]$ and $f_{Irrad}(\lambda)[\mathrm{uW/cm^2/nm/DN}]$ are the radiance and irradiance calibration functions, respectively; $f_{Rad}(\lambda)$ and $f_{Irrad}(\lambda)$ also depend on viewing geometry and solar incident angles, respectively, but those arguments are left out for brevity. The EMI-2 instrument BSDF $BSDF(\lambda)[\mathrm{sr^{-1}}]$ is defined as follows:

$$BSDF(\lambda) = \frac{f_{Rad}(\lambda)}{f_{Irrad}(\lambda)} \tag{3}$$

On-orbit earth reflectance spectrum $R(\lambda)$ is the ratio of atmospheric radiance $L_{Earth\text{-}rad}(\lambda)$ and solar irradiance $E_{Sun\text{-}irrad}(\lambda)$:

$$R(\lambda) = \frac{\pi L_{Earth\text{-}rad}(\lambda)}{\mu \cdot E_{Sun\text{-}irrad}(\lambda)} \tag{4}$$

where $\mu = \cos\theta$ is the cosine of the solar zenith angle $\theta$. From Equations (1)–(4), $R(\lambda)$ can be expressed as follows:

$$R(\lambda) = \frac{\pi}{\mu} \cdot \frac{S_{Earth}}{S_{Sun}} \cdot BSDF(\lambda) \tag{5}$$

Equation (5) shows that reflectance spectrum $R(\lambda)$ can be directly obtained from the EMI-2 instrument $BSDF_{EMI\text{-}2}(\lambda)$.

The EMI-2 instrument forward model of Earth and Sun mode are described as follows:

$$S_{Earth}(\lambda) = L_{Earth}(\lambda) \cdot \tau_{Earth}(\lambda) \cdot \eta_d(\lambda) \tag{6}$$

$$S_{Sun}(\lambda) = E_{Sun}(\lambda) \cdot BRDF(\lambda) \cdot \tau_{Sun}(\lambda) \cdot \eta_d(\lambda) \tag{7}$$

where $\tau_{Earth}(\lambda)$ and $\tau_{Sun}(\lambda)$ are the optical transmission of the earth and sun path, respectively, and the differences between these two optical paths are determined by primary mirror in Earth mode and solar mesh and folding mirror in Sun mode; $BRDF(\lambda)$ is diffuser bidirectional reflectance distribution function (BRDF) characterization, and $\eta_d(\lambda)$ is the

response parameter of the EMI-2 detector system. From Equations (1), (2), (7) and (8), the following can be obtained:

$$f_{Rad}(\lambda) = \frac{1}{\tau_{Earth}(\lambda) \cdot \eta_d(\lambda)} \tag{8}$$

$$f_{Irrad}(\lambda) = \frac{1}{BRDF(\lambda) \cdot \tau_{Sun}(\lambda) \cdot \eta_d(\lambda)} \tag{9}$$

From Equations (3), (6), and (7), EMI-2 instrument BSDF can be expressed as follows:

$$BSDF_{EMI\text{-}2}(\lambda) = BRDF(\lambda) \cdot \tau(\lambda) \tag{10}$$

where $\tau(\lambda) = \frac{\tau_{Sun}(\lambda)}{\tau_{Earth}(\lambda)}$. Equation (10) shows that EMI-2 instrument BSDF is mainly determined by the primary telescope mirror in Earth mode and solar mesh, onboard diffuser, and folding mirror in Sun mode. The most important contributor for EMI-2 instrument BSDF is the onboard diffuser.

### 2.1. Absolute Irradiance Calibration

A 1000 W FEL quartz tungsten halogen lamp, which has been calibrated by NIST, is used for EMI-2 absolute irradiance calibration. EMI-2 absolute irradiance calibration function represents the conversion relationship between output digital number (DN) and input irradiance ($uW/cm^2/nm$). The lamp is moved on a rail to change the distance to the onboard diffuser. The EMI-2 instrument is mounted on a 2D turntable to change the lamp incident angles. The absolute irradiance calibration system is shown in Figure 2.

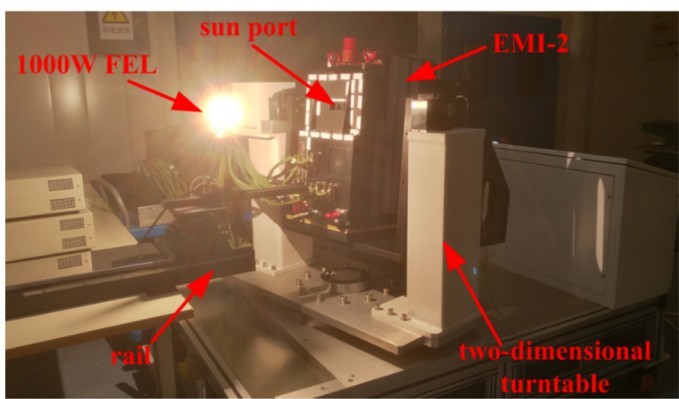

**Figure 2.** Absolute irradiance calibration system of EMI-2.

The details of absolute irradiance calibration process are as follows:

A laser is used to align the center of the lamp and the onboard diffuser, the lamp is moved to three positions on a rail at 30, 40, and 50 cm from the diffuser, which are labelled as $d_1, d_2$, and $d_3$, respectively.

The turntable drives the EMI-2 to rotate in the horizontal and vertical directions, which can cover the on-orbit solar incident angles. At each rotation angle, 20 lamp images $S_{Sun}(\lambda)$ are continuously collected. All measurements are performed at three lamp positions.

Irradiance calibration functions, calculated from the three distances, are averaged to obtain the final result. Irradiance calibration function $f_{Irrad}(\lambda)$ is calculated by as follows:

$$f_{Irrad}(\lambda) = \frac{E_{Diff}(\lambda)}{S_{Sun}(\lambda)} \tag{11}$$

In Equation (11), lamp images $S_{Sun}(\lambda)$ are corrected by the background measurement, $E_{Diff}(\lambda)$ is the irradiance on the diffuser. The EMI-2 diffusers are 46 mm in length and 16 mm in width, the imaging area is 44 mm × 2 mm, and the nominal angle between the

incident light and the diffuser normal is 22°. Here the lamp is considered a point source, the distances between the lamp and spots on the diffuser are different, and the irradiance on one diffuser spot is different from others due to the distance offset and inverse-squared law. Distance offset $\Delta d(\theta_i, d_0, \theta_v)$ is determined by lamp incident angle $\theta_i$, distance from lamp to diffuser center (corresponding to the center row) $d_0$, and viewing angle $(-57° \sim 57°)$ $\theta_v$. $E_{Diff}(\lambda)$ in Equation (11) is obtained by the following:

$$E_{Diff}(\lambda) = (\frac{d_0 + \Delta d}{50})^2 \cdot E_{NIST\text{-}50cm}(\lambda) \tag{12}$$

where $E_{NIST\text{-}50cm}(\lambda)$ is the lamp irradiance, which has been calibrated by NIST at a distance of 50 cm.

### 2.2. Absolute Radiance Calibration

A large aperture integrating sphere system with eight 200 W tungsten halogen lamps is used for EMI-2 absolute radiance calibration, and the output radiance levels are measured by a spectroradiometer with absolute accuracy traceable to the NIST. The EMI-2 absolute radiance calibration function represents the conversion relationship between output DN and input radiance (uW/cm$^2$/sr/nm). EMI-2 has to be rotated in six steps to complete the total viewing field of 114° because approximately 20° can be illuminated once. The absolute radiance calibration system for EMI-2 is shown in Figure 3.

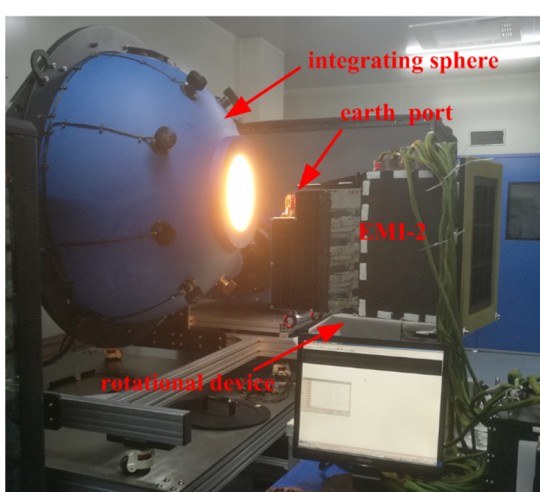

**Figure 3.** Absolute radiance calibration system of EMI-2.

The details of absolute radiance calibration are as follows:

EMI-2 is placed facing the integrating sphere exit aperture, and its position is adjusted such that the center of the exit aperture can be imaged in the central field of view of EMI-2. By moving the EMI-2 with a rotational device, all swath angles can be calibrated one by one.

Different output radiance levels for EMI-2 calibration are selected. One hundred response images are measured by EMI-2 at each illuminated angle, and background measurements of each channel are collected.

The response images after deduction of background are averaged to obtain mean value $S_{Earth}(\lambda)$. Output radiance $L(\lambda)$ is measured by the spectroradiometer. Radiance calibration function $f_{Rad}(\lambda)$ is derived.

$$f_{Rad}(\lambda) = \frac{L(\lambda)}{S_{Earth}(\lambda)} \tag{13}$$

*2.3. On-Ground Measurement*

In order to evaluate the pre-launch radiometric characterization of EMI-2, an on-ground measurement is performed. EMI-2 is placed in clean room with a quartz window inlayed in the wall, and receives the scattered sunlight passing through the window. The sun port observes the direct sunlight through a plane mirror, the precise value of the reflectivity is not known, see Figure 4.

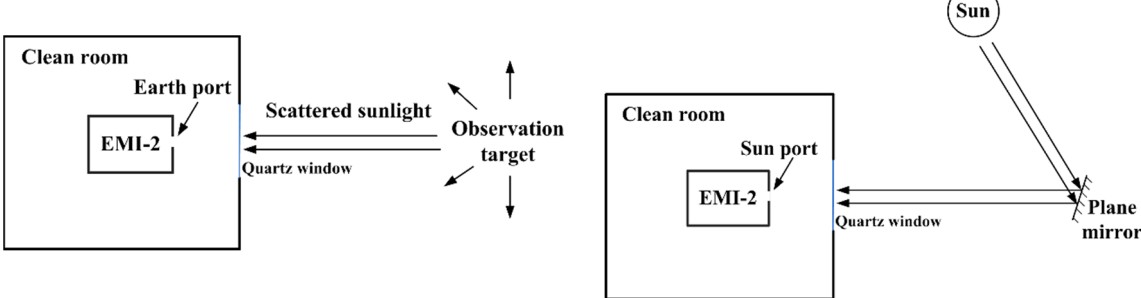

**Figure 4.** Schematic of EMI-2 experimental setup. The observation target includes walls, fences, weeds, and tree trunks.

The absolute irradiance calibration functions $f_{Irrad}(\lambda)$, radiance calibration functions $f_{Rad}(\lambda)$ and instrument BSDF are applied to the on-ground measurements. Observation target radiance $L(\lambda)$ and direct solar irradiance $I_{Sun}(\lambda)$ are calculated by

$$L(\lambda) = f_{Rad}(\lambda) \cdot DN_{Earth}(\lambda), \tag{14}$$

$$I_{Sun}(\lambda) = f_{Irrad}(\lambda) \cdot DN_{Sun}(\lambda), \tag{15}$$

where $DN_{Earth}(\lambda)$ and $DN_{Sun}(\lambda)$ are output digital number of Earth and Sun ports, respectively.

## 3. Results

Based on Equation (10), the BSDF parameters can be calculated theoretically. For EMI-2, two onboard solar diffusers (SD), one F4 diffuser and one quartz volume diffuser (QVD) diffuser, are adopted for solar measurement. QVD is used frequently to provide a solar reference spectrum, and F4 is used on a long-time basis to monitor QVD degradation.

F4 and QVD diffuser BRDF are characterized during on-ground calibration. BRDF calibration angles are determined by in-orbit solar incident angles on diffusers and installation position of diffusers. Diffuser BRDF results are shown in Figure 5.

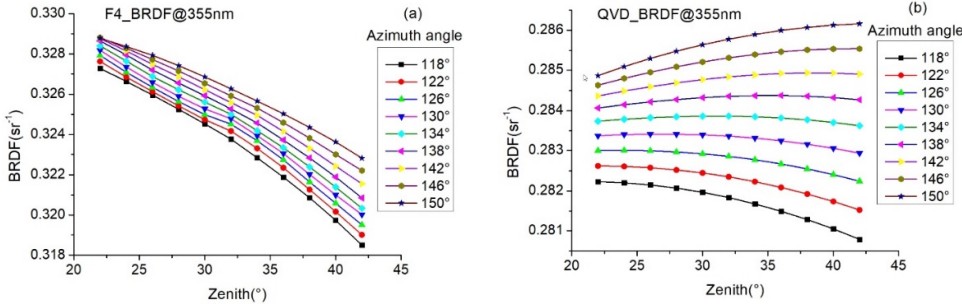

**Figure 5.** Onboard diffuser BRDF at 355 nm: (**a**) F4 and (**b**) QVD with the zenith angle and azimuth angle of incident light varying from 22°–44° and 118°–150°, respectively.

An NIST-calibrated 1000 W FEL quartz tungsten halogen lamp and an external spectralon plate (ESP) with known BRDF are selected to determine $\tau(\lambda)$ parameters. The ESP and internal F4 diffusers are illuminated by the FEL lamp with the same distance between the FEL lamp and the diffusers. The $\tau(\lambda)$ parameters are shown in Figure 6.

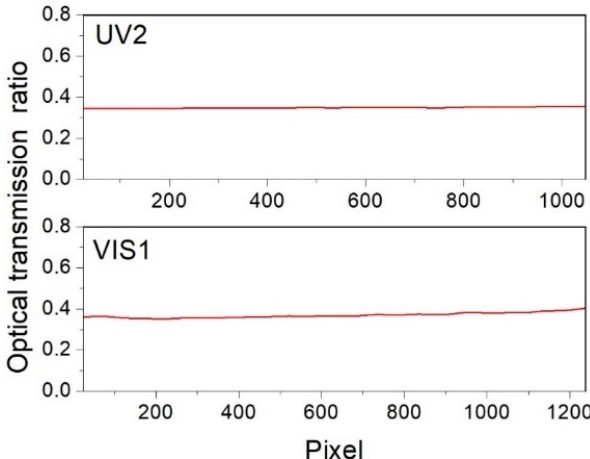

**Figure 6.** Optical transmission ratio of sun and earth paths.

The results in Figure 4 show that $\tau(\lambda)$ is approximately 0.35. Based on solar diffuser BRDF and $\tau(\lambda)$ parameters, EMI-2 BSDF is approximately $0.11[\text{sr}^{-1}]$. The exact value of EMI-2 BSDF is obtained from absolute irradiance and radiance calibration.

*3.1. EMI-2 Wavelength*

Spectral calibration sources are used to illuminate a diffuser in Sun mode, which can achieve wavelength calibration of the full field of view. Wavelength maps for EMI-2 are shown in Figure 7.

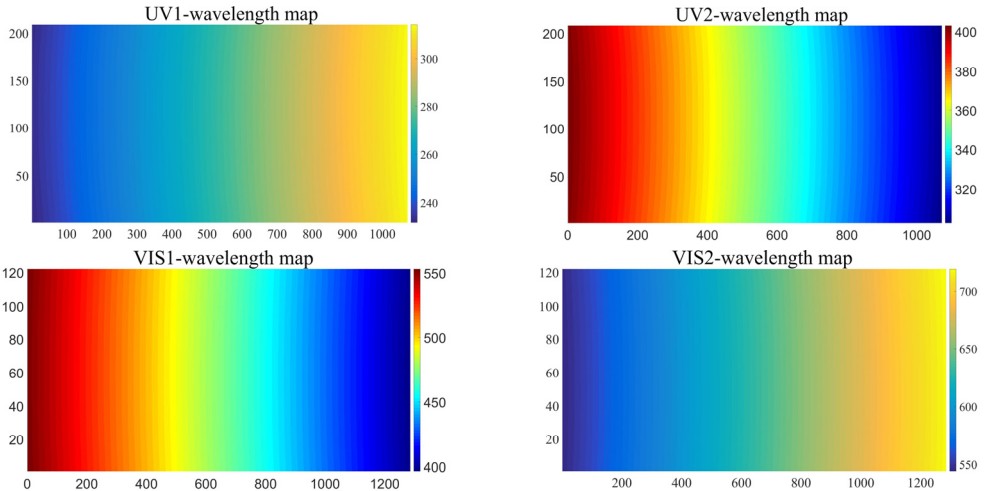

**Figure 7.** Wavelength maps for UV1, UV2, VIS1, and VIS2 channels, where horizontal and vertical directions are spectral and spatial dimensions, respectively. The spectral smile (the wavelengths at the center pixels of imaging spectrometer detector array are different from the marginal pixels) is visible in wavelength maps.

The EMI-2 spectral response can be described by a Gaussian-type function. The full width at half maximum (FWHM) is obtained by Gaussian fitting (see Table 2).

**Table 2.** FWHM of EMI-2 spectral response.

| FOV | UV1/nm | UV2/nm | VIS1/nm | VIS2/nm |
|---|---|---|---|---|
| 50° | 0.39 | 0.45 | 0.47 | 0.48 |
| 40° | 0.37 | 0.46 | 0.42 | 0.48 |
| 30° | 0.38 | 0.47 | 0.41 | 0.49 |
| 20° | 0.38 | 0.48 | 0.40 | 0.50 |
| 10° | 0.39 | 0.48 | 0.40 | 0.50 |
| 0° | 0.40 | 0.49 | 0.41 | 0.49 |
| −10° | 0.41 | 0.49 | 0.42 | 0.48 |
| −20° | 0.40 | 0.49 | 0.44 | 0.46 |
| −30° | 0.40 | 0.48 | 0.46 | 0.44 |
| −40° | 0.39 | 0.47 | 0.50 | 0.41 |
| −50° | 0.43 | 0.48 | 0.58 | 0.39 |

### 3.2. EMI-2 Instrument BSDF

For absolute irradiance calibration, the calculated distance offsets are shown in Figure 8.

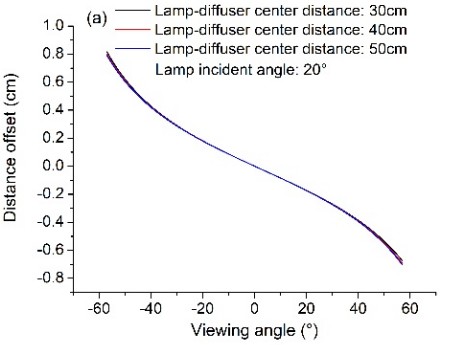 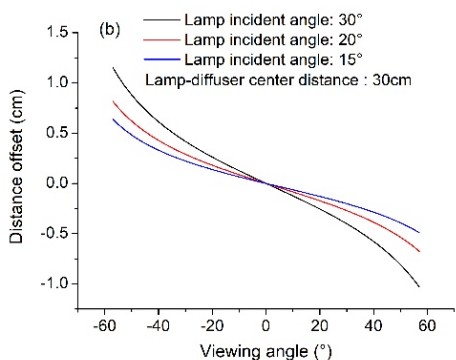

**Figure 8.** Calculated distance offsets. (**a**) at lamp incident angle 20° with distance from lamp to diffuser center 30, 40, and 50 cm; (**b**) at distance 30 cm with lamp incident angles 30°, 20°, and 15°.

The results in Figure 8 show that lamp incident angle $\theta_i$ is the major influencing factor. The lamp images via F4 and QVD diffuser are shown in Figure 9.

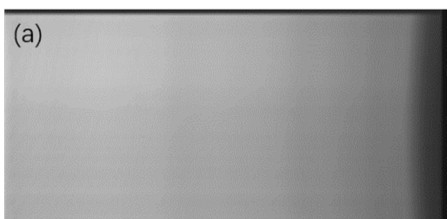 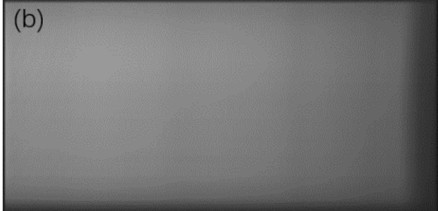

**Figure 9.** Lamp images via F4 (**a**) and QVD (**b**) diffuser in VIS1 channel, where horizontal and vertical directions are spectral and spatial dimensions, respectively.

The EMI-2 absolute irradiance calibration function results are shown in Figure 10.

The EMI-2 absolute radiance calibration function results are shown in Figure 11.

Based on absolute radiance and irradiance calibration functions, EMI-2 instrument BSDF is calculated by Equation (3). The results are shown in Figure 12.

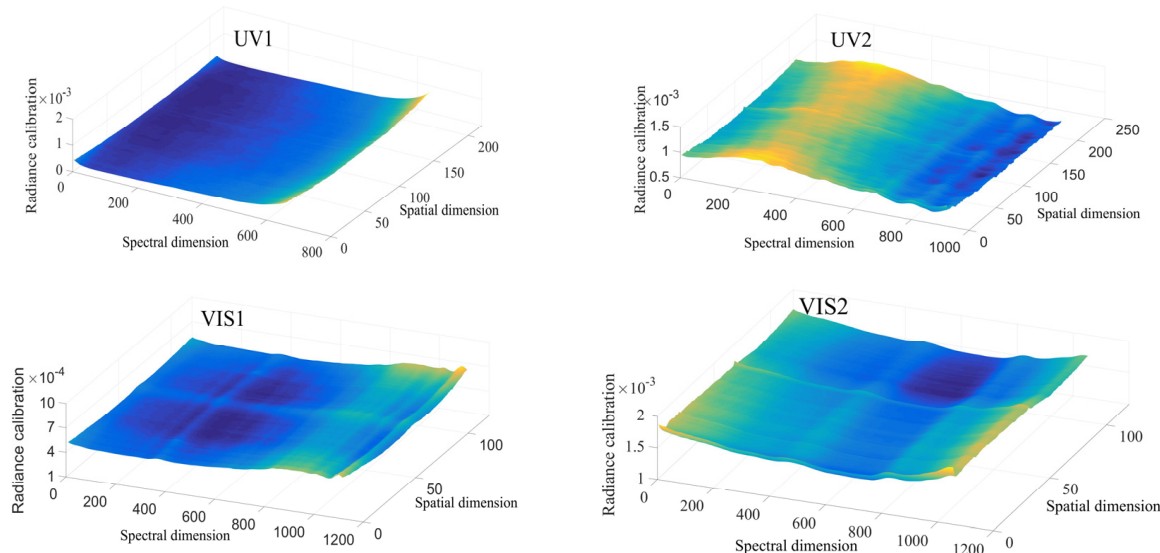

**Figure 10.** Absolute irradiance calibration function results for the nominal azimuth (22°) and elevation (0°) angles.

**Figure 11.** Absolute irradiance calibration function results for the nominal azimuth (22°) and elevation (0°) angles.

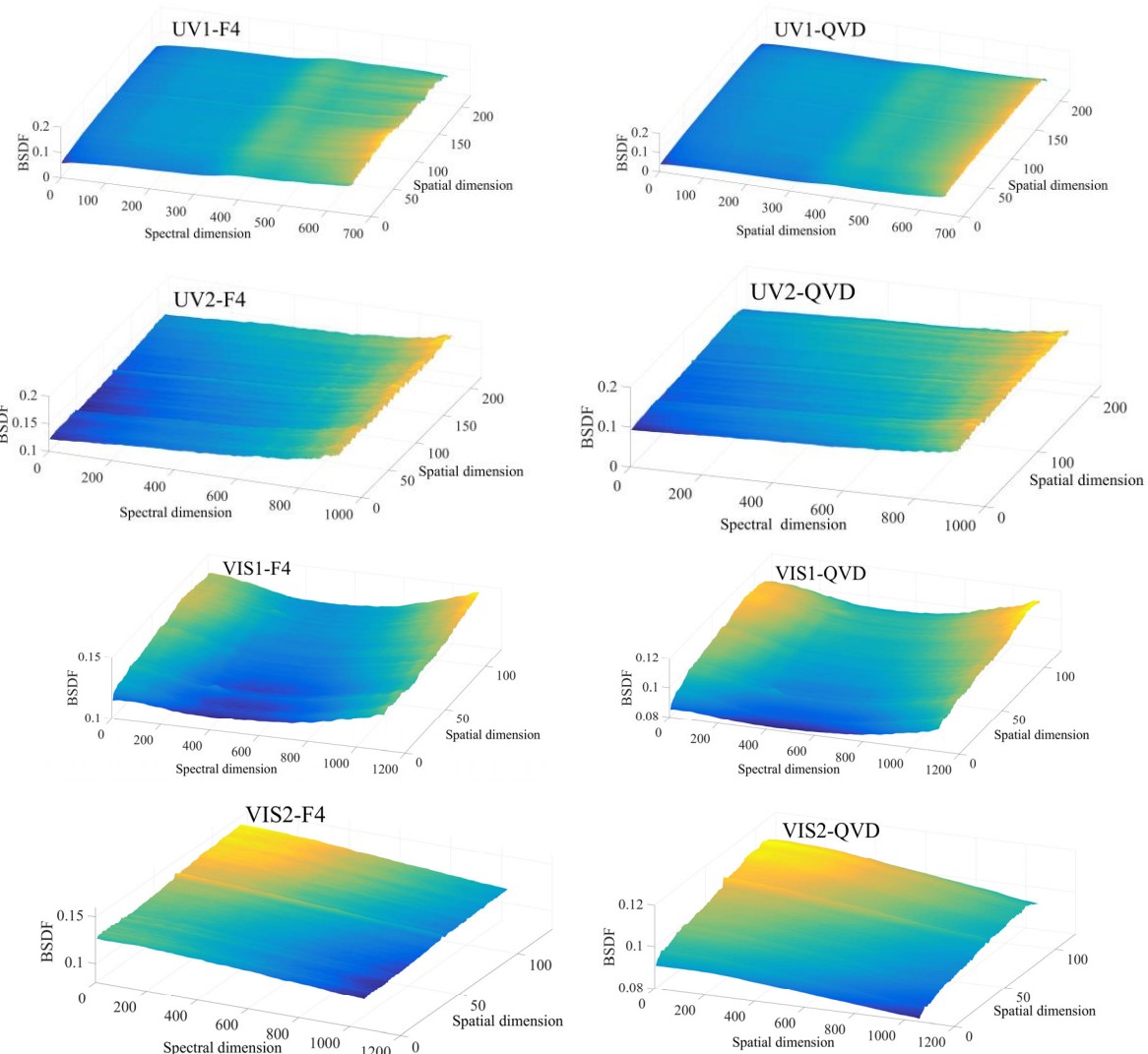

**Figure 12.** EMI-2 instrument BSDF.

The EMI-2 instrument BSDF is a function of wavelength (spectral dimension), viewing direction (spatial dimension), and incident angles of the onboard diffuser.

### 3.3. Uncertainty

The uncertainty of EMI-2 instrument BSDF is mainly determined by irradiance calibration system accuracy, radiance calibration system accuracy, and nonlinearity and nonstability response of EMI-2. Irradiance calibration system uncertainty $\mu_i$ depends mainly on the lamp calibrated by NIST. Radiance calibration system uncertainty $\mu_r$ refers to spectroradiometer and integrating sphere system. The spectroradiometer has been calibrated by a standard diffuser plate and an NIST-calibrated lamp. The nonlinearity $\mu_l$ of the EMI-2 can be expressed as follows:

$$\mu_l = \frac{\delta}{\overline{S}} \tag{16}$$

where $\delta$ is the standard deviation of linear fit residuals, and $\overline{S}$ is the mean value of the pixel response. The linear fitting results of EMI-2 are shown in Figure 13.

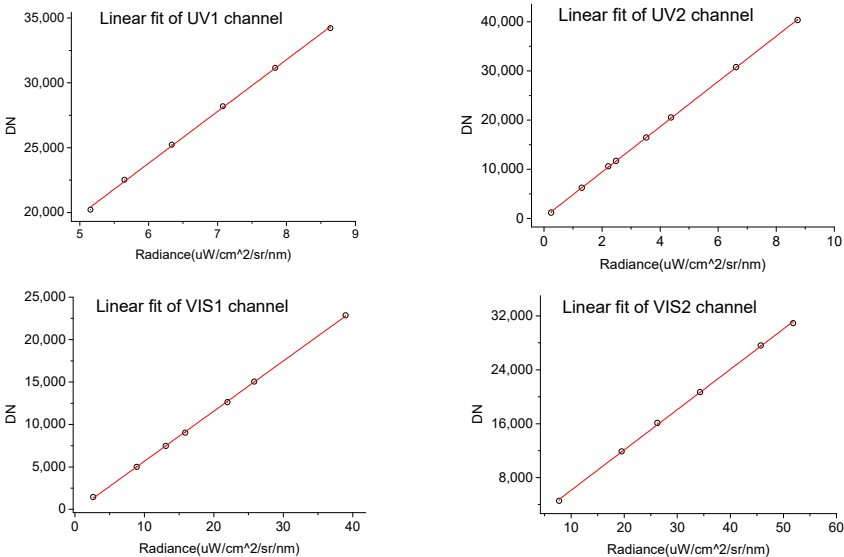

**Figure 13.** Linear fitting results of EMI-2 response.

The nonstability $\mu_s$ of the EMI-2 can be expressed as follows:

$$\mu_s = \frac{\sqrt{\frac{\sum\limits_{i=1}^{n}(S_s - \overline{S}_s)^2}{n-1}}}{\overline{S}_s} \tag{17}$$

where $n$ is the total number of measurements, $S_s$ is the $i$th response of the pixel, and $\overline{S}_s$ is the mean response of the total measurements. Irradiance and radiance calibration function uncertainty depend on nonlinearity and nonstability. The uncertainty $\mu$ of EMI-2 instrument BSDF can be calculated by the following:

$$\mu = \sqrt{\mu_i^2 + \mu_r^2 + 2\mu_l^2 + 2\mu_s^2} \tag{18}$$

The results of EMI-2 instrument BSDF uncertainty are shown in Table 3. EMI-2 adopts DOAS retrieval technique, DOAS is sensitive to the spectral calibration, the spectral stability and spectrally dependent features of the instrument [7]. The EMI calibration uncertainty is about 5%, the NO2 and total ozone columns retrieval results show that a 5% error of calibration is enough for the application.

**Table 3.** EMI-2 instrument BSDF uncertainty.

| Uncertainty Factor | Channels | | | |
|---|---|---|---|---|
| | UV1 | UV2 | VIS1 | VIS2 |
| Irradiance calibration system (%) | 2.8 | 2.1 | 1.7 | 1.7 |
| Radiance calibration system (%) | 4.0 | 3.6 | 3.6 | 3.6 |
| Nonlinearity (%) | 0.4 | 0.7 | 0.8 | 1.0 |
| Nonstability (%) | 0.3 | 0.3 | 0.1 | 0.1 |
| Combined uncertainty (%) | 4.9 | 4.3 | 4.1 | 4.2 |

*3.4. On-Ground Measurement*

The scattered sunlight radiance and direct sunlight irradiance spectra measured by EMI-2 are shown in Figure 14.

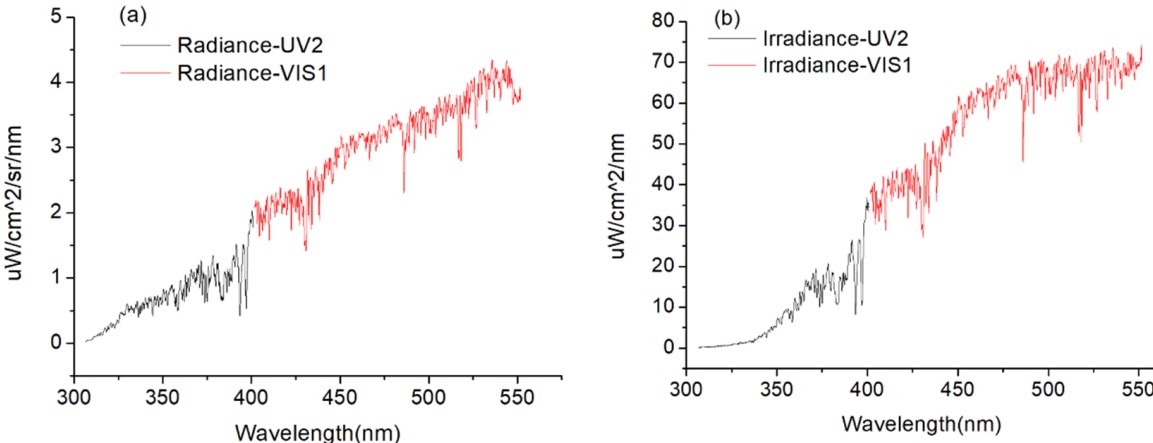

**Figure 14.** Radiance (**a**) and irradiance (**b**) measured by EMI-2. Note that quartz window transmittance and plane mirror reflectivity are involved in the measurements.

Assuming observation target is Lambertian scattering, EMI-2 instrument BSDF parameters are used to calculate the target reflectance spectrum by Equation (5). The results are shown in Figure 15.

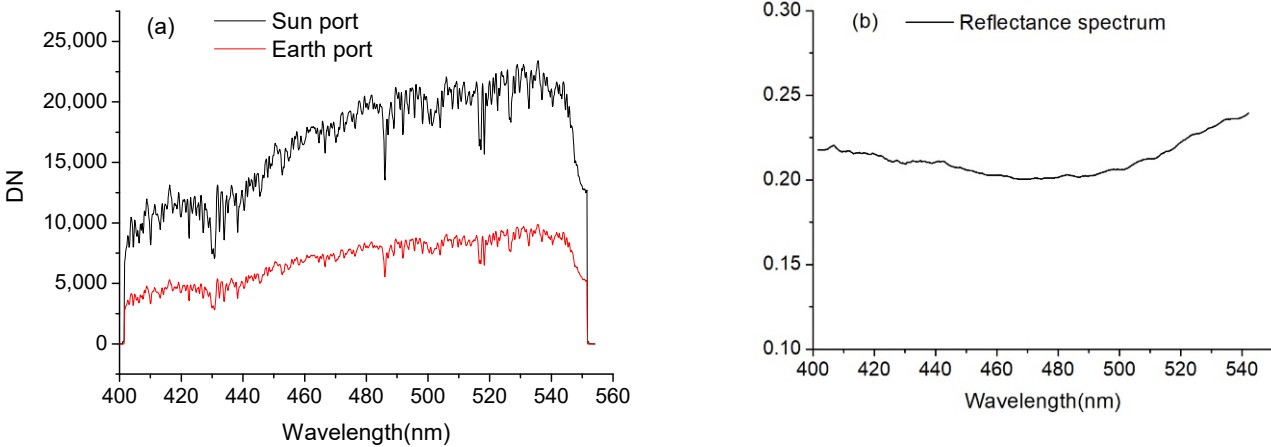

**Figure 15.** (**a**): Output digital number of Earth and Sun ports, (**b**): observation target reflectance spectrum after smoothing. Note that quartz window transmittance and plane mirror reflectivity are involved in the measurements.

The on-ground measurement results show that EMI-2 instrument BSDF parameters can be used to calculate the reflectance spectrum, the on-orbit application method is discussed in next section.

## 4. Discussion

Pre-Launch EMI-2 instrument BSDF parameters are obtained on-ground, the on-orbit application are as follows:

QVD and F4 are planned to be used weekly and monthly, respectively. The EMI-2 instrument BSDF parameters need to be interpolated to match the on-orbit solar incident angle intervals. The F4 diffuser is mainly used as reference solar diffuser to monitor the QVD degradation.

The solar images $S_{Sun}(\lambda, \theta_i, \varphi_i)$ observed via QVD during a solar observation sequence are selected, solar incident azimuth $\varphi_i$ and zenith $\theta_i$ angles are calculated, and then the EMI-2 instrument BSDF $BSDF(\lambda, \theta_i, \varphi_i)$ at the corresponding angles is determined. Approximately 95 solar images are collected during a solar observation sequence of 150 s, solar zenith angle $\theta_i$ varies from +5° to −5°, and the zenith angles are corrected to normal

angle $\theta_i = 0$ by the goniometry correction factor. Moreover, the averaged solar images $\overline{S_{Sun}(\lambda, \theta_i = 0, \varphi_i)}$ are obtained to improve the signal-to-noise ratio, and the corrected and averaged instrument BSDF is $\overline{BSDF(\lambda, \theta_i = 0, \varphi_i)}$. Based on Equation (5), earth reflectance spectrum $R(\lambda)$ can be calculated from digital number $S_{Earth}(\lambda)$ of the earth path by the following:

$$R(\lambda) = \frac{\pi}{\mu} \cdot \frac{S_{Earth}(\lambda)}{S_{Sun}(\lambda, \theta_i = 0, \varphi_i)} \cdot \overline{BSDF(\lambda, \theta_i = 0, \varphi_i)} \tag{19}$$

Instrument BSDF degradation is calculated. On-orbit signals of earth and sun optical paths are time dependent, and the two paths' degradation are different. As discussed above in the EMI-2 instrument BSDF, most of the optical components are common to the earth and sun optical paths, and the optical sensitivity of these common components cancels in the ratio of absolute irradiance and radiance calibration. The material of the primary telescope mirror is the same as that of the folding mirror. Therefore, instrument BSDF degradation is mainly caused by the degradation of solar mesh and diffuser. This degradation can be monitored by F4 diffuser, and the ratio $\gamma(\lambda, t)$ of QVD and F4 instrument BSDF can be expressed as follows:

$$\gamma(\lambda, t) = \frac{BSDF_{QVD}(\lambda, t)}{BSDF_{F4}(\lambda, t)} \tag{20}$$

F4 is used on a long-time basis and well protected; hence, $BSDF_{F4}(\lambda, t)$ does not change with time $t$. From Equations (2) and (3), the following can be obtained:

$$\gamma(\lambda, t) = \frac{S_{Sun\text{-}F4}(\lambda, t)}{S_{Sun\text{-}QVD}(\lambda, t)} \tag{21}$$

where $S_{Sun\text{-}QVD}(\lambda, t), S_{Sun\text{-}F4}(\lambda, t)$ are output digital number of Sun mode via QVD and F4 at on-orbit time $t$, respectively. The QVD instrument BSDF degradation $\alpha(\lambda, t)$ can be calculated as follows:

$$\alpha(\lambda, t) = \gamma^*(\lambda, t) - \gamma(\lambda, t) \tag{22}$$

where $\gamma^*(\lambda, t)$ is the ratio of output digital number via F4 and QVD at the first time of on-orbit solar observations.

## 5. Conclusions

EMI-2 instrument BSDF is obtained by preflight calibration and mainly used to calculate earth reflectance spectrum. Based on EMI-2 on-orbit solar incident angles, the instrument F4 and QVD BSDF look-up tables are established. F4 is much less frequently used to minimize potential degradation and exposure time, which is adopted to update the on-orbit QVD BSDF look-up tables. EMI-2 instrument BSDF is used as key input parameter for the L1b data processor to produce on-orbit earth reflectance spectrum products because most of the optical components' on-orbit degradation can be cancelled, and the earth reflectance spectrum calculated accuracy of EMI-2 is improved compared with EMI. Moreover, EMI-2 instrument BSDF can improve the knowledge of instrument radiometric response.

**Author Contributions:** Conceptualization, F.S. and W.L.; validation, C.J.; resources, Y.J. and S.W.; data curation, K.Z.; writing—original draft preparation, M.Z.; writing—review and editing, H.Z. and M.Z. All authors have read and agreed to the published version of the manuscript.

**Funding:** This research was funded by National Natural Science Foundation of China, grant number 41705016.

**Institutional Review Board Statement:** Not applicable.

**Informed Consent Statement:** Not applicable.

**Data Availability Statement:** The data presented in this study are available on request from the corresponding author.

**Acknowledgments:** The authors would like to thank the reviewers for their precious comments and suggestions.

**Conflicts of Interest:** The authors declare no conflict of interest.

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
