# Peer review of "Pre-Launch Radiometric Characterization of EMI-2 on the GaoFen-5 Series of Satellites"

_remotesensing, doi:10.3390/rs13142843_

Round 1
Reviewer 1 Report
Pre-Launch Radiometric Characterization of EMI-2 on the 2 GaoFen-5 series of satellites 3
MinJie Zhao et al.
The article describes the pre-launch calibration of the EMI-2 instrument. It is important for the users to know something of the calibration of the instrument data that they are using, e.g. how it relates to international standards, and to appreciate the likely stability of the instrument after launch. The article is well written, with generally good use of English. As far as I can tell, the principles used are sound and the article is clearly written.
Some minor suggestions:
Line 29: It is explained in the Abstract that EMI-2 is for launch on GeoFen-5(02) but this should be mentioned in the text also.
Comment: There is no mention of GF-5(02) or EMI-2 in WMO’s OSCAR database https://space.oscar.wmo.int/satellites/view/gf_5. It’s a pity that CNSA appear not to notify WMO of future missions.
Line 49: Spell out DOAS –differential optical absorption spectroscopy. Reference?
Figure 1: what is the meaning of “len”?
Line 169: “The response images deducted by background”, better English would be “The response images after deduction of background”
Figure 4 and Fig 16: What is the observation target physically?
Line 177: It is mentioned that the mirror has “unknown reflectivity”. This seems a bit misleading, as it is a mirror so reflectivity is presumably close to 1. You could perhaps say “the precise value of the reflectivity is not known”, or similar.
Fig 5: Is this figure actually telling us anything? I just see two circular disks on a black background.
Line 220 (fig 8 caption): what is meant by “spectral smile”?
Fig 11, 12 & 13: font is very small on the axes. Can anything be done to improve? Also fig 8.
Fig 16: How can we know whether the reflectances are reasonable? See earlier comment about not being told what the target is.
General: It is stated that EMI-2 calibration is an improvement compared with EMI, and that the EMI BSDF has not been determined on ground. Is there a way to quantify this improvement, in terms of the impact on parameters that are of interest to the user?
Author Response
The authors would like to thank the reviewer’s precious comments and suggestions. Please see the attachment.

Reviewer 2 Report
The Chinese Gaofen series satellite offers unique capabilities for a better observation of the earth environment from different aspects, the Gaofen 5 is a new attempts to get the atmospheric components, the paper introduces the calibration process and evaluation of the expected calibration uncertainties, the paper is interesting for the community, I have a few comments as below
(1) As the authors mentioned, the instrument is similar to GOME, OMI, and TROPOMI, it will be interesting to see some example of the comparison with those instruments over different atmospheric conditions
(2) How the stray light issue is dealt with in Gaofen 5
(3) Will be a 5% error of calibration enough for the application, some statement is needed.
(4) According to the wavelength settings and calibration investigation, what other atmospheric parameters will be provided by Gaofen5 besides the typical NO2, SO2, O3
Author Response

(The authors gave the same response as above.)
